# A Computer-Aided Diagnostic System for Diabetic Retinopathy Based on Local and Global Extracted Features

Sayed Haggag [1], Ahmed Elnakib [2], Ahmed Sharafeldeen [2], Mohamed Elsharkawy [2], Fahmi Khalifa [2], Rania Kamel Farag [3], Mohamed A. Mohamed [1], Harpal Singh Sandhu [2], Wathiq Mansoor [4], Ashraf Sewelam [3] and Ayman El-Baz [2],*

1   Electronics and Communications Department, Faculty of Engineering, Mansoura University, Mansoura 35516, Egypt
2   Bioengineering Department, University of Louisville, Louisville, KY 40292, USA
3   Ophthalmology Department, Faculty of Medicine, Mansoura University, Mansoura 35516, Egypt
4   College of Engineering and Information Technology, University of Dubai, Dubai P.O. Box 14143, United Arab Emirates
*   Correspondence: aselba01@louisville.edu; Tel.: +1-502-468-9248

**Featured Application: This paper presents a novel deep learning system for the detection and diagnosis of diabetic retinopathy using optical coherence tomography images.**

**Abstract:** Diabetic retinopathy (DR) is a major public health problem and the leading cause of vision loss in the working age population. This paper presents a novel deep learning system for the detection and diagnosis of DR using optical coherence tomography (OCT) images. The input for this system is three-channel local and global information from OCT images. The local high-level information is represented by the thickness channel and the reflectivity channel. The global low-level information is represented by the grey-level OCT original image. The deep learning system processes the three-channel input to produce the final DR diagnoses. Experimental results on 200 OCT images, augmented to 800 images, which are collected by the University of Louisville, show high system performance related to other competing methods. Moreover, 10-fold and leave-one-subject-out (LOSO) experiments are performed to confirm how significant using the fused images is in improving the performance of the diagnoses, by investigating four different CNN architectures. All of the four architectures achieve acceptable performance and confirm a significant performance improvement using the fused images. Using LOSO, the best network performance has improved from 90.1 ± 2% using only the grey level dataset to 97.7 ± 0.5% using the proposed fused dataset. These results confirm the promise of using the proposed system for the detection of DR using OCT images.

**Keywords:** diabetic retinopathy; deep learning; low level information; high level information

## 1. Introduction

Diabetic retinopathy is a major public health problem and the leading cause of vision loss in the working age population. DR occurs when the blood vessels in the retina are damaged due to high blood sugar. These damaged blood vessels can cause blurry vision or stop the blood flow. Advanced DR causes vision loss. DR usually affects both eyes [1]. In 2015, worldwide, around 415 million people have diabetes [2]. This number is expected to reach 642 million by 2040 [2]. As this number increases, so too will the number of DR cases. Records show that early detection and diagnosis can lead to the protection of eyesight. Noninvasive medical image modalities have been used to detect and diagnose DR. The most popular image modalities for the detection and diagnosis of DR are fundus imaging and optical coherence tomography (OCT) imaging. Visual diagnosis using these modalities is subjective and time consuming. Therefore, automated methods [1,3–11] have

been presented in the literature for the automated detection and diagnosis of DR. The aim of this paper is to present an automated platform for the detection and diagnosis of DR.

Different methods in the literature have been proposed for the automated diagnosis of DR. The most popular categories of the related work, based on the imaging modality, are fundus imaging and OCT imaging. For example, Bhardwaj et al. [4] presented a hierarchical system for the detection and grading of DR. They tested their system on both Messidor and IDRiD fundus datasets, achieving a 94% accuracy on IDRiD dataset. Qiao et al. [1] used a semantic deep learning network to detect DR from fundus images. Wang et al. [7] used a semisupervised multichannel generative adversial network to grade fundus images. Gadekallu et al. [8] used principal component analysis (PCA) for data reduction followed by a grey wolf optimization to choose the optimal hyperparameters to train a deep learning model. The goal of this model is to predict DR using fundus images. Oh et al. [10] used a deep learning DR system to early detect DR from ulta-wide field fundus images. Akram et al. [12] presented a four-step system for the detection and classification of DR from fundus imaging. The system was composed of preprocessing, candidate lesions detection, the formulation of the feature set, and classification. The detection of candidate lesions extracted any region that might contain a lesion using a filter bank. The feature set was based on shape, intensity, and statistics to classify the previously detected regions as either normal, mild, moderate, or severe DR. Wan et al. [13] used transfer learning and hyperparameter tuning to train AlexNet, VggNet, GoogleNet, and ResNet on Kaggle fundus imaging datset. The best classification accuracy was 95.68%. Shankar et al. [14] classified fundus images to various DR severity levels based on a three-stage system: preprocessing, histogram-based segmentation, and a synergic deep learning model. The system was tested on the Messidor DR dataset. Zhang et al. [15] combined the popular convolutional neural networks (CNN) with the customized standard deep neural networks to classify fundus images into several grades. Pratt et al. [16] used data augmentation and CNN to classify fundus images. Gao et al. [17] used a CNN to grade fundus images, achieving an accuracy of 88.72% for a four-grade classification task. Sayres et al. [18] showed that a deep learning algorithm, using fundus imaging, can improve the accuracy of DR manual diagnosis. Zhao et al. [19] used a deep learning model that integrated an attention feature extraction model with a bilinear fine-grained classification model. This deep learning model was trained using a log softmax loss function in order to grade fundus images. Li et al. [20] used a deep learning model based on attention modules for joint DR and diabetic macular edema grading using fundus images. Gulshan et al. [21] compared a deep learning algorithm with manual grading, showing that the deep method either equals or exceeds manual grading on an Indian fundus dataset.

On the other hand, reports emphasize the OCT's high optical capabilities and high DR diagnostic accuracy, which are independent of depth or spatial resolution [11]. OCT has been repeatedly used for DR detection and diagnosis. For example, Sandhu et al. [22] presented a computer aided diagnostic (CAD) system for DR diagnostic and grading using OCT images. The system segmented the retina into 12 layers. Then, they estimated the reflectivity, curvature, and thickness of each layer. These features were fed to a neural network that distinguished between normal retinas and those with nonproliferative DR. Further, the system graded the level of the DR. Li et al. [23] used two deep networks for the early detection of DR from OCT images. One network was used to extract features from the original OCT images, and the other network was used to extract features from OCT retinal layer segmentation. A classification block was used to integrate the features extracted from both networks. ElTanboly et al. [24] segmented the OCT image into 12 layers. Then, they extracted the reflectivity, curvature, and thickness features from each layer. A deep network is trained on the cumulative probability distribution function (CDF) of the extracted features. Experiments, run on 40 training and 12 test OCT scans, achieved a detection accuracy of 92%. Sandhu et al. [25] trained a classifier for automated DR detection from OCT-angiography (OCTA) based on three types of features: blood vessel density, blood vessel calibre, and the size of foveal avascular zone. Their system achieved

an accuracy of 94.3%. More recently, Sandhu et al. [26] extracted three features from each OCT layer: reflectivity, thickness, and curvature. Their developed system extracted four features from OCT angiography (OCTA): blood vessel caliber, vessel density, the size of the foveal avascular zone, and the number of bifurcation and crossover points. A random forest classifier is used for the diagnosis and grading of DR based on the combined features from OCT and OCTA. Alam et al. [27] used 120 OCTA images and trained a supported vector machine on six extracted features: blood vessel density, blood vessel tortuosity, blood vascular caliber, vessel perimeter index, foveal avascular zone area, and foveal avascular zone contour irregularity. Combined feature classification achieved an accuracy of 94.41% for control vs. DR and an accuracy of 92.96% for control vs. mild NPDR stage. The main disadvantages of the previous work are as follows:

- Some studies contain a low number of test subjects;
- Traditionally, the input to the CNN is the original image, which may not be sufficient to report a high accuracy score;
- The overall accuracy can be improved.

The main motivation behind this work is to achieve better accuracy by importing important features related to the disease. Unlike the work in the literature, the current study presents a novel fusion of features. Two medical markers are taken into account: the reflectivity and thickness features. Each of these features is represented by an input channel to a deep convolutional neural network (CNN). Three channels representing the high-level information (thickness and reflectivity) and low-level information (grey level image) are input to the CNN network. The main contributions/features of the proposed work are as follows:

- The novel representation of the three input channels of the CNN network to integrate local (high-level) and global (low-level) information of the OCT image;
- The overall accuracy is improved compared to the related work.

The rest of this paper is organized as follows. Section 2 represents the materials and methods. Section 3 illustrates the experimentation and their discussions. Section 4 is the discussion. Section 5 concludes the paper.

## 2. Materials and Methods

The proposed CAD system to diagnose DR disease, based on the central B-scan (through the foveal pit) of the volumetric OCT scans, is illustrated in Figure 1. This proposed system consists of three steps. (i) The segmentation of the twelve retinal layers from the OCT B-scan using a previously developed appearance-based approach. (ii) Descriptive markers are extracted, including the higher-order reflectivity metric, in addition to morphological features (i.e., thicknesses) from each segmented layer. A three-channel input is formed from a reflectivity map, a thickness map, and the OCT image to act as an input for a diagnostic CNN. (iii) A diagnostic CNN is used to determine the final diagnosis of the B-scan as normal or DR. In this section, we will illustrate in detail each of these steps.

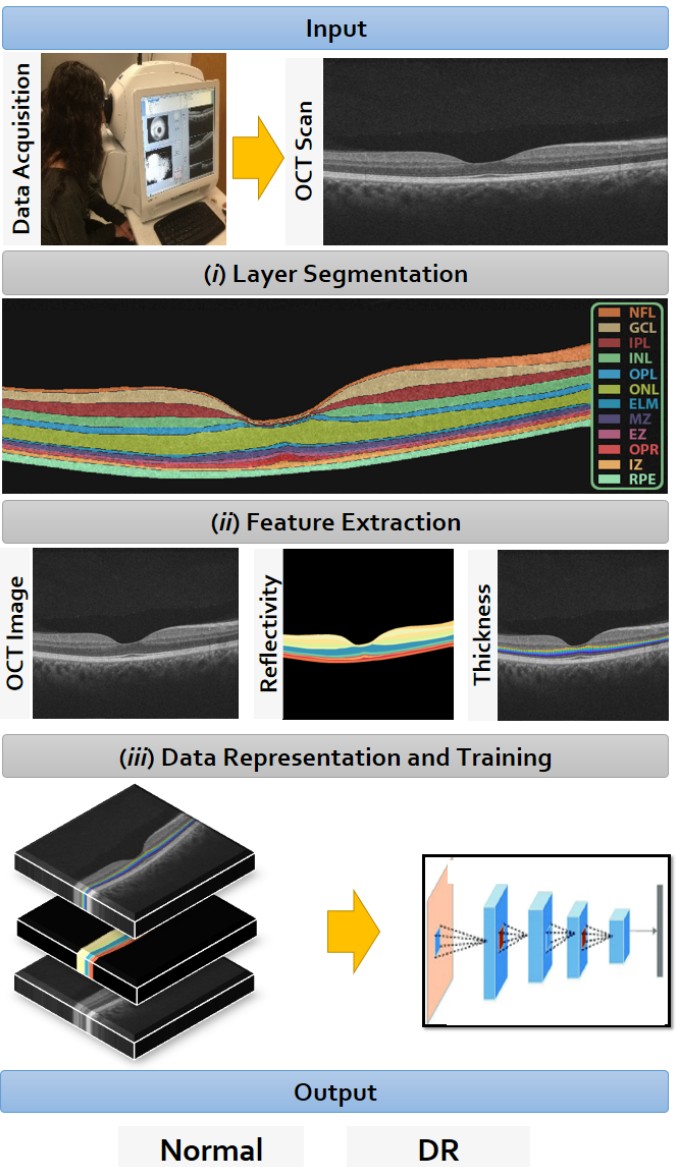

**Figure 1.** Proposed CAD system for DR diagnostic using OCT.

### 2.1. Data Collection

The proposed framework was trained and tested using OCT data, acquired utilizing a Zeiss Cirrus HD-OCT 5000 machine. The machine is located at the Department of Ophthalmology and Visual Science, University of Louisville. The machine has an axial resolution of 5 μm. University of Louisville's Institutional Review Board (IRB) certifies the data collection and acquisition. The study adhered to the Declaration of Helsinki. The collected dataset includes 200 OCT images from 100 subjects (i.e., two eyes per subject, one scan for each eye). The database is balanced and contains 100 OCT normal images and 100 DR images. The image resolution is 1024 × 1024.

### 2.2. Retinal Layers Segmentation

The proposed CAD system starts with layer segmentation. In this step, the OCT image is segmented into 12 layers using a previously developed appearance-based approach (Figure 2) [24]. The output is the twelve-layer segmentation of the OCT image. These layers are the nerve fiber layer (NFL), ganglion cell layer (GCL), inner plexiform layer (IPL), inner nuclear layer (INL), outer plexiform layer (OPL), outer nuclear layer (ONL), external

limiting membrane (ELM), myoid zone (MZ), ellipsoid zone (EZ), outer photoreceptor segments (OPR), interdigitation zone (IZ), and retinal pigment epithelium (RPE).

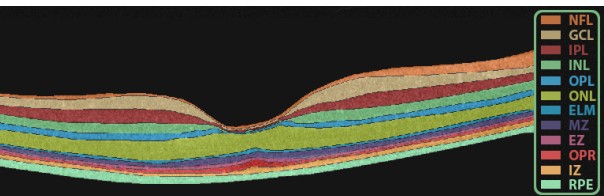

**Figure 2.** Segmentation of the retinal layers.

*2.3. CNN Input Data Representation*

One of the contributions of this work is the novel representation of the input of the utilized deep CNN. Unlike the traditional representation of inputting the OCT image, herein, we used a novel input CNN data representation, composed of three channels (Figure 3). The first channel is the low-level information of the OCT image after removing the background (BG). The second channel is the high-level processed information of the reflectivity map, represented as the average retinal layer's gray level that is assigned to each layer. The third channel is the high-level processed information of the thickness map, calculated as the thickness of each region between each two successive retinal layers. A 3-channel fused image is composed of three image maps: an OCT image without the BG, the reflectivity map, and the thickness map. Below, we will illustrate each of these channels.

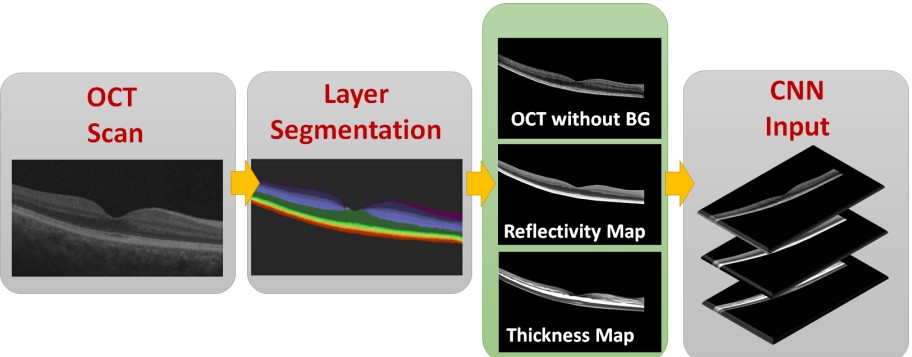

**Figure 3.** Data representation of the input of the CNN.

2.3.1. OCT without BG

By applying the 12-layer segmentation as a mask, the OCT image without the BG can be found. After removing the BG, the OCT image is normalized by the following equation [28]:

$$I_n = \frac{I_{in} - RV}{R_{12} - RV} \times 1000 \qquad (1)$$

where $I_n$ is the normalized grey level intensity; $I_{in}$ is the input pixel's grey level intensity; and $RV$ and $R_{12}$ are the average intensity of the vitreous and the average intensity of the RPE layer, respectively. Before estimating the reflectivity, a given OCT B-scan is first normalized by the given equation as the OCT pixel gray level is not an absolute metric of reflectivity since it depends on some external factors, such as pupil dilation, that affect image quality. For example, the retinal NFL in an eye that is insufficiently dilated may appear darker than in a fully dilated eye, even in the case where both eyes are of the same age and free of pathology. Therefore, a relative metric of reflectivity is used, where the reflectivity of the NFL and other layers is a fraction of the RPE reflectivity. It is standardized with respect to the RPE because that layer is typically preserved in early DR [29].

### 2.3.2. Reflectivity Map

The reflectivity map is composed of two steps. First, the reflectivity feature is calculated for each layer as the mean of the reflectivity of this layer (the mean intensity of the reflection of the light of a retinal layer). Second, the mean reflectivity is assigned to each layer of the 12-segmented layers. The reflectivity of a given layer *i* is defined as:

$$Ref_i = mean(\text{grey levels of the layer } i) \tag{2}$$

Figure 4 shows the reflectivity maps for a normal subject (Figure 4a) vs. a DR subject (Figure 4b), normalized between 0 and 1 for visualization. Comparing the normal and DR in the figure: (i) the comparable layers have different reflectivity. In addition, (ii) forming a bigger layer makes differences between the reflectivity at the same locations. These remarks suggest that the reflectivity map can contribute to distinguishing normal and DR subjects.

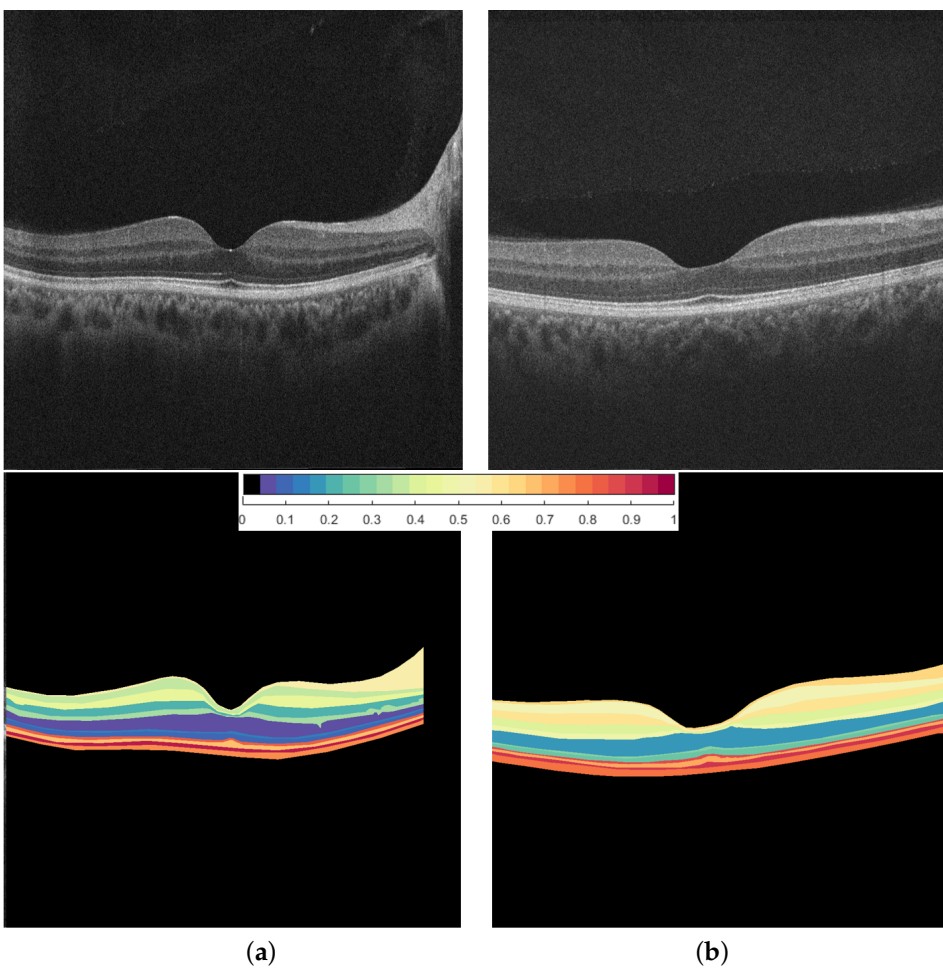

| (**a**) | (**b**) |

**Figure 4.** Reflectivity maps for (**a**) a normal subject and (**b**) a DR subject. First row is the original images and second row is the normalized colored reflectivity maps for visualization.

### 2.3.3. Thickness Map

The thickness map is formulated by extracting the thickness feature over the 12 layers of the OCT image. Each layer is divided into 143 regions. In each region out of 143 regions, the thickness feature is assigned to this region, formulating the thickness map. The thickness is measured as the Euclidean distance between two correspondence points on the two layer boundaries. To allocate point correspondence, a geometrical approach is applied based on solving the Laplace equation between the two boundaries (see Figures 5 and 6). Let $\gamma$ be

a scalar field that represents the electric field between the two layers. The Laplace equation is defined as

$$\nabla^2 \gamma = \frac{\partial^2 \gamma}{\partial x^2} + \frac{\partial^2 \gamma}{\partial y^2} = 0 \tag{3}$$

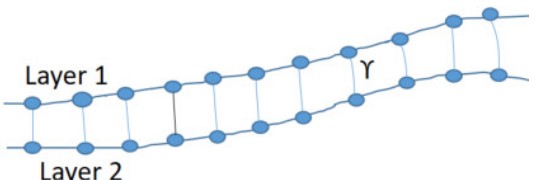

**Figure 5.** Estimation of the thickness between each of the two layers of the retina.

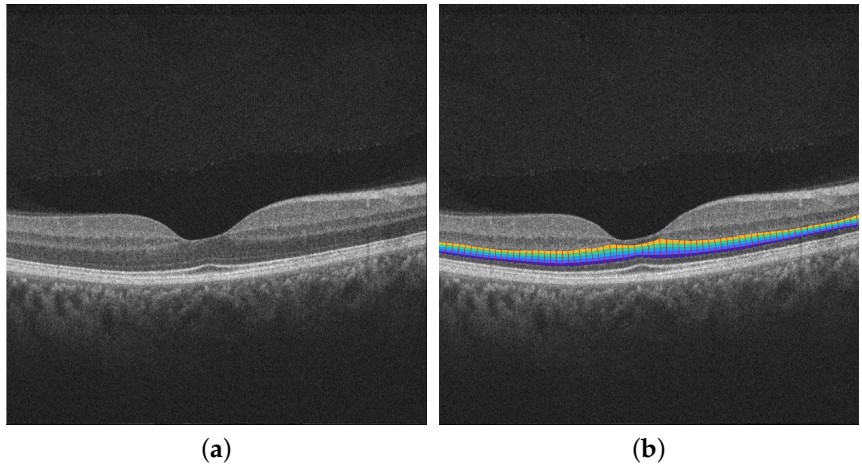

**Figure 6.** Original image (**a**) and the colored thickness map (**b**) between two retinal layers.

To estimate $\gamma(x, y)$, the iterative Jacobi approach is used:

$$\gamma^{i+1}(x,y) = 1/4\{\gamma^i(x + \Delta x, y) + \gamma^i(x - \Delta x, y) + \gamma^i(x, y + \Delta y) + \gamma^i(x, y - \Delta y)\} \tag{4}$$

where $\gamma^i(x, y)$ is the electric field at $(x, y)$ on the $i$th iteration and $\Delta x$ and $\Delta y$ are the step resolutions on $x$ and $y$ directions, respectively.

Figure 7 shows the thickness maps for a normal subject (Figure 7a) vs. a DR subject (Figure 7b). As shown in the figure, some of the layers have different thicknesses. In addition, forming a bigger layer produces differences between the thickness of normal vs. DR subjects at the same locations. These remarks suggest that the thickness map can contribute efficiently to distinguishing between normal and DR subjects.

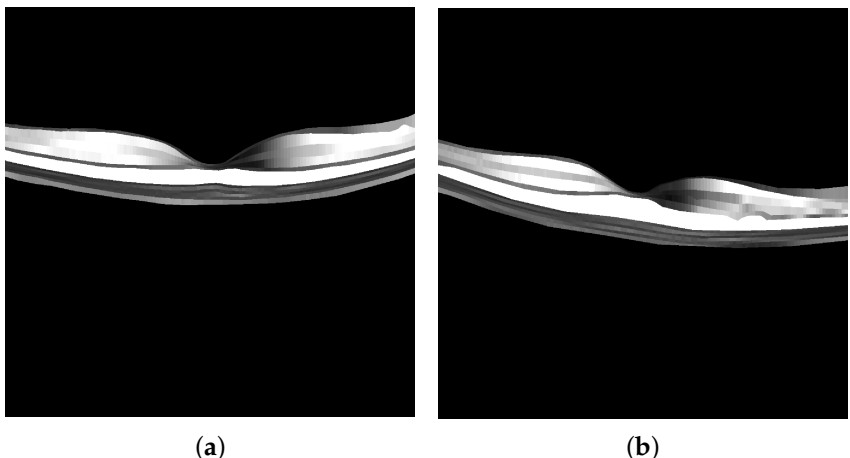

**(a)**  **(b)**

**Figure 7.** Thickness maps for (**a**) a normal subject and (**b**) a DR subject.

### 2.4. Data Representation to the CNN

The original dataset contains 100 normal OCT images and 100 diabetic retinopathy images. All of them are of the size (1024 × 1024). For each image, there is a reflectivity map (1024 × 1024) composed of 28 parts for 12 layers. Additionally, there is a thickness map (1024 × 1024). The thickness map and the reflectivity map are added up to the original gray level image to compose a fused image. Now, we have two datasets: the original gray level dataset and the fused images dataset. Each data set, after augmentation, is used separately to train the CNN to compare the effect of fused images in improving the classification capability. All the images in both datasets are scaled to (256 × 256) to fit the input size of the proposed CNN.

### 2.5. Diagnostic CNN

The goal of this step is to detect and diagnose DR (i.e., classify the image to either normal or DR). Since the database size is not too large, we adopt both transfer learning and learning from scratch. AlexNet [30] is learned from scratch, while transfer learning is applied for DneseNet121 [31] and ResNet101 [32]. Alexnet is composed of eight layers (five convolutional layers and three fully connected layers). Moreover, DenseNet121 and ResNet101 are adopted for transfer learning after removing the top layer and replacing it with a fully connected layer with dropout 40% and a final node of sigmoid activation function for classification. These networks are pre-trained on the well-known "Imagenet" dataset [33].

In addition to the standard CNNs, we built a simple network to be faster and easier to train. We carried out many experiments to optimize the hyperparameters of this CNN. The results led to the final structure composed of four convolutional layers, two fully connected layers, and a final binary node for classifying normal vs DR. We called this network "DRNet":

- First convolutional layer with input of 256 × 256 and 16 kernels;
- Max. pooling layer;
- Second convolutional layer with 32 kernels;
- Max. pooling layer;
- Third convolutional layer with 64 kernels;
- Max. pooling layer;
- Fourth convolutional layer with 128 kernels;
- Max. pooling layer;
- The output of the last layer is then flattened and applied to the fully connected NN input;
- First layer of a fully connected dense layer of 4096 nodes with ReLu activation function;

- Second dense layer of 1024 nodes with ReLu activation function;
- Final single node with sigmoid activation to discriminate the two classes.

All networks have been tested on the available database to investigate their abilities for the detection and diagnosis of DR.

## 3. Results

Many experiments are carried out to optimize the hyperparameters of the networks. For AlexNet, the optimization results estimate the learning rate to be 0.001, and the momentum = 0.9. The loss function is the binary cross entropy using the ADAMv2 optimizer. DRNet uses the same learning rate and momentum with the ADM optimizer (from the keras library). On the other hand, the DenseNet121 and ResNet101 training phase is optimized to a 0.0005 learning rate and the momentum = 0.9. The loss function is also binary cross entropy using the Adam optimizer.

Here, we confirm that we use interpretable, explainable AI, which means that humans can understand the decision made by AI by importing the important features related to the disease. In our case, this is done by adding the reflectivity and thickness to the original images to reinforce the capability of diagnosis [34].

### 3.1. Proposed System Results

It is noted that the testing accuracy is limited despite the high accuracy through the training phase, which indicates overfitting in the training phase. To mitigate the effect of this drawback, we used the technique of dropping out a portion of the fully connected network. Many experiments have been done to deduce the optimal ratio of the drop out portion, and it is found to be 40%. It is noted that this value is critical, and the validation accuracy is very sensitive to this value. This technique has improved the testing accuracy by at least 6%.

As a comparison, the results of the classification accuracy, sensitivity, and specificity of the trained models are reported in Table 1. The table contains both fused and original gray image dataset classification results. The performance of DRNet showed an acceptable performance compared to the other models, especially in the fused images dataset.

As a first trial in the experiments, we divided each dataset ten-fold, taking into account that both eye images of the same person are either in the training folds only or in the testing fold only to avoid overfitting. Data augmentation and transfer learning are also used to overcome the possible overfitting. The data are augmented by horizontal flipping and random rotation, which results in an increase of data size to 800 images. After that, we applied DRNet, as well as other models, separately. The classification metrics are computed for each fold and then averaged. The gray-level image results and the comparison with the fused images are shown in Table 1. These results are the average percentage and the standard deviation (STD) of many experiments. It is clear from Table 1 that the STDs in case of gray images, especially in the case of the AlexNet results, are very large, even after 1000 epochs of training, which is an indicator of the instability of the performance of AlexNet in the gray-level dataset. One advantage of DRNet is the stability of performance despite the low accuracy in the case of gray-level images compared to that of AlexNet.

On the other hand, using our proposed fused-image dataset has greatly increased the accuracy and other metrics as well as reduced the STD in using either DRNet or other models, as shown in Table 1. It is clear that the accuracy is increased by an average of about 20%, and the STD is decreased by about one-fifth in the results of AlexNet and DRNet. Moreover, sensitivity and specificity are improved by an average of 16%.

Due to the limited number of images, it is suggested to use leave-one-subject-out (LOSO) in the training phase to adjust more realistic representative results. The results are shown in Table 2 for the four used CNN models. It is clear from these results that the performance of the DRNet is very close to that of the AlexNet. On the other hand, the results show the superior performance of the proposed fused images technique in all networks. In the training phase, AlexNet takes about double or triple the number of training epochs

more than that required for DRNet. Moreover, it takes also about double the time required to complete a single epoch, which in turn means that the required time to complete one training phase of AlexNet is about five times or greater than its corresponding time in the case of DRNet. The transfer learning-based networks converge at a lower number of training epochs but with a longer time for each individual epoch.

The comparison and performance evaluation of the different used CNN models can be investigated using the ROC curves and the area under the curve (AUC). Each model is evaluated separately on the fused-images dataset and the results are shown in (Figure 8). From this figure, we can conclude that AlexNet and Densenet121 have the best performance.

**Table 1.** Ten-fold accuracy (Acc), sensitivity (Sen), and specificity (Sp) comparisons for fused images and the gray level images trained over different models (all values are percentiles).

| Used Network | Gray Acc | Gray Sen | Gray Sp | Fused Acc | Fused Sen | Fused Sp |
|---|---|---|---|---|---|---|
| DRNet | $76.3 \pm 1.4$ | $76.0 \pm 1.0$ | $77.7 \pm 0.9$ | $93.3 \pm 1.4$ | $92.5 \pm 0.6$ | $94.0 \pm 0.9$ |
| AlexNet | $73.8 \pm 8$ | $72.0 \pm 5$ | $75.9 \pm 2.7$ | $93.5 \pm 2$ | $93.1 \pm 1.2$ | $93.5 \pm 0.6$ |
| DenseNet121 | $89.8 \pm 3$ | $86.1 \pm 3$ | $92.4 \pm 2.2$ | $94.8 \pm 2$ | $96.0 \pm 5$ | $94.5 \pm 0.7$ |
| ResNet101 | $82.8 \pm 6$ | $82.1 \pm 5$ | $86.2 \pm 4.7$ | $88.5 \pm 2$ | $85.0 \pm 5$ | $91.2 \pm 4.7$ |

**Table 2.** LOSO accuracy, sensitivity, and specificity comparisons for both graylevel and fused images (all values are percentiles).

| Used Network | Gray Acc | Gray Sen | Gray Sp | Fused Acc | Fused Sen | Fused Sp |
|---|---|---|---|---|---|---|
| DRNet | $85.3 \pm 5$ | $83.1 \pm 1.1$ | $86.6 \pm 0.9$ | $97.1 \pm 0.7$ | $98.0 \pm 0.4$ | $96.7 \pm 0.5$ |
| AlexNet | $86.1 \pm 3$ | $84.0 \pm 2.9$ | $89.0 \pm 3.5$ | $97.7 \pm 0.5$ | $98.1 \pm 0.6$ | $98.3 \pm 0.5$ |
| DenseNet121 | $90.1 \pm 2$ | $87.8 \pm 2$ | $93.1 \pm 2.4$ | $97.3 \pm 0.4$ | $98.3 \pm 0.5$ | $98.0 \pm 0.4$ |
| ResNet101 | $83.2 \pm 3$ | $84.3 \pm 4.1$ | $86.1 \pm 3.7$ | $91.6 \pm 2.8$ | $92.3 \pm 2.1$ | $91.1 \pm 1.8$ |

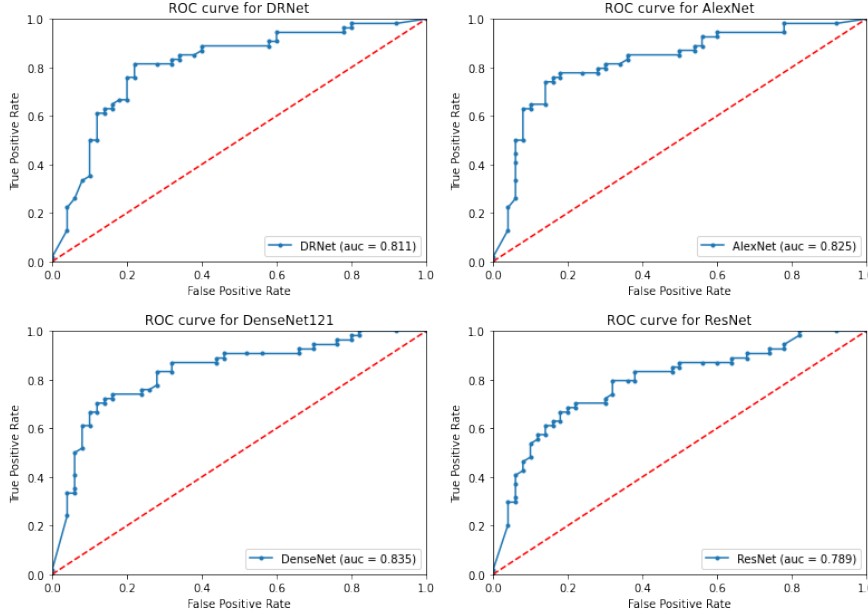

**Figure 8.** The ROC curves for the different four CNN models to compare and evaluate these models.

### 3.2. Ablation Study

To confirm the effect of using fused images in our proposal, we added an ablation study. In this study, two datasets are produced and applied to the same deep learning

models that are used in the previous experiments. The first dataset contains images that are composed of three layers: the gray level in the first layer, the reflectivity map in the second layer, and the third layer is a replica of the second layer. The other dataset is similar, but the reflectivity map is replaced by the thickness map.

Each dataset is divided into 10 folds. The accuracy and other metrics are computed for each fold when the training is applied to the other nine folds. The operation is repeated for each fold in either dataset. The results of deep models experiments are shown in Tables 3 and 4. These experiments are carried out to study the effect of adding both reflectivity map and thickness map to the gray level images in improving the classification metrics. The results shown in Table 4 investigate the effect of adding only the thickness amp, which results in great improvements in accuracy compared to the results shown in Table 1 for gray images. On the other hand, the accuracy is slightly less than the results shown in Table 1 for fused images, which confirms the importance of adding both maps of thickness and reflectivity to the gray level images. The same conclusion about the effect of adding the reflectivity map only can be drawn by noting the results shown in Table 3 and comparing it to Table 1.

A statistical comparison between the results, with the proposed technique and without it, is carried out using the two-sample Student's *t*-test. The obtained *p*-values illustrate that there is a statistically significant difference between the results ($p$-value $\leq 0.05$). These results indicate that adding both the reflectivity map and the thickness map to the gray level images statistically improves the classification accuracy.

**Table 3.** Ten-fold comparisons for images composed of gray level and the reflectivity map only.

| Applied Network | Acc% | Sen% | Sp% |
|---|---|---|---|
| DRNet | $88.1 \pm 1.5$ | $84.0 \pm 2.2$ | $90.0 \pm 1.7$ |
| AlexNet | $89.5 \pm 2.8$ | $86.1 \pm 2.5$ | $92.1 \pm 2.7$ |
| DenseNet121 | $89.3 \pm 3$ | $86.2 \pm 1.6$ | $92.5 \pm 1.8$ |
| ResNet101 | $82.2 \pm 6$ | $80.3 \pm 3.1$ | $85.0 \pm 2.9$ |

**Table 4.** Ten-fold comparisons for images composed of gray level and thickness map only.

| Applied Network | Acc% | Sen% | Sp% |
|---|---|---|---|
| DRNet | $89.9 \pm 0.8$ | $86.0 \pm 2.2$ | $92.6.1 \pm 2.0$ |
| AlexNet | $91.8 \pm 1.4$ | $89.0 \pm 2.9$ | $94.5 \pm 2.1$ |
| DenseNet121 | $91.3 \pm 2$ | $88.0 \pm 1.8$ | $94.4 \pm 2.3$ |
| ResNet101 | $84.1 \pm 5$ | $82.3 \pm 4$ | $85.1 \pm 3.1$ |

## 4. Discussion

The reported results in Section 3.1 describe the contribution of adding the thickness and reflectivity maps to the original low-level images, which achieves an accuracy ranging from 97.1 to 97.7% (Table 2). In addition, each map is added separately to the gray level image, as shown in Section 3.2 (the ablation study), and the results summary is reported in Tables 3 and 4.

As a comparison with related work, we list the accuracies attained by different techniques in Table 5. The comparison also reported the accuracy of the related works on the same data.

**Table 5.** Related work accuracy comparison.

| Related Work | Dataset | Accuracy% |
|---|---|---|
| Proposed System | same dataset | 97.1 |
| Sandhu et al. [25] | same dataset | 94.3 |
| ElTanboly et al. [24] | same dataset | 92.0 |
| Alam et al. [27] | different dataset | 93.5 |
| Gao et al. [17] | different dataset | 88.7 |

From the above results, we can conclude the merit of using fused images in improving the classification capability of the CNN compared to the usual original images. Extracting important medical features, such as thickness and reflectivity here, and adding them to the original images greatly improves the diagnosis accuracy, sensitivity, and specificity, which in turn reinforces the idea of using fused images, especially in medical image classification and automatic diagnosis. This may be due to the difficulty of classifying medical images based on appearance only, which is a neglected aspect of other distinguishing features and signs of a particular disease, which are not sufficiently visible in the usual images. Using explainable AI through the fusion of medical diagnostic markers greatly enhances performance using either learning from scratch or transfer learning. Alexnet, which learns from scratch, and Densenet, which uses transfer learning, achieve the best accuracies, while Resnet and DRnet achieve lower but acceptable accuracies due to the overfitting of ResNet (contains many layers) and the underfitting of DRNet, which contains a low number of layers.

## 5. Conclusions

This paper presents a new deep method for DR detection and diagnosis. The main novelty of the proposed method is the fusion of the low-level information (grey level image) with the high-level information (reflectivity and thickness images) using a three-channel data representation to the input of the deep CNN. Different CNNs are investigated, and the superiority of the proposed fusion is indicated using the three-channel data representation over using the traditional image. The average accuracy in the case of low-level gray images is found to be 86%, with an upper limit of 90.1%. However, in the case of high-level fused images, the accuracy is improved to an average of 97.5%. The great improvement in accuracy as well as sensitivity and specificity supports using this technique in DR objective detection, especially when comparing these results with the results of related work. These preliminary results come with the cost of the effort required to extract the reflectivity and thickness maps. In the future, we will investigate using other deep CNN networks. In addition, more data will be tested in order to investigate the reliability of the proposed method.

**Author Contributions:** S.H., A.E., F.K. and A.E.-B.: conceptualization and formal analysis. S.H., A.S. (Ahmed Sharafeldeen), M.E. and A.E.-B.: methodology. S.H., A.S. (Ahmed Sharafeldeen) and M.E.: software development. S.H., A.E., A.S. (Ahmed Sharafeldeen), M.E. and A.E.-B.: validation and visualization. S.H., A.E., F.K., R.K.F. and A.E.-B.: initial draft. R.K.F., M.A.M., A.S. (Ashraf Sewelam) and A.E.-B.: resources, data collection, and data curation. S.H., A.E., H.S.S. and A.E.-B.: review and editing. A.E., W.M. and A.E.-B.: project administration. M.A.M., A.S. (Ashraf Sewelam), W.M. and A.E.-B.: project directors. All authors have read and agreed to the published version of the manuscript.

**Funding:** This research received external funding from the Academy of Scientific Research and Technology (ASRT) in Egypt (Project No. JESOR 5246).

**Institutional Review Board Statement:** The study was conducted according to the guidelines of the Declaration of Helsinki and approved by the Institutional Review Board of University of Louisville (protocol code 18.0010 and date of approval 24 January 2019).

**Informed Consent Statement:** Informed consent was obtained from all subjects involved in the study.

**Data Availability Statement:** Data are available upon reasonable request to the corresponding author.

**Conflicts of Interest:** The authors declare no conflict of interest.

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
