# Peer review of "A Computer-Aided Diagnostic System for Diabetic Retinopathy Based on Local and Global Extracted Features"

_applsci, doi:10.3390/app12168326_

Round 1
Reviewer 1 Report
A new deep learning method for DR problem is presented. The novelty is based on the introduction of preprocessed info (thickness, brightness and reflectivity).
Authors show that the introduction of this info in the input tensor improves the detection rate. They work on two different architectures.
My major concern is about the way authors show the results. They do not compare with other methods (only in the discussion section; this way is too complicated to compare to; it is important that authors run the other algorithms in the same conditions, so we can assure that the improvement is due to the new layers)
An statistical analysis of results is missing. The results seem too similar to say there is a real statistically significant difference.
Since all this new info is extracted from original images using linear operations, I do not understand why the results improve. CNNs are able to extract the features (even in strong nonlinear spaces) so authors must provide an explanation. Is it just because the input tensor is larger so CNN is learning more parameters? In other words, it is because of the network or beacuse of the input signal? Some experiments with simpler architectures can be run and give some light.
The paper has some typos such as repeated words "the the", "Rreflectivity" and so on.
Author Response
|
1) Authors show that the introduction of this info in the input tensor improves the detection rate. They work on two different architectures. My major concern is about the way authors show the results. They do not compare with other methods (only in the discussion section; this way is too complicated to compare to; it is important that authors run the other algorithms in the same conditions, so we can assure that the improvement is due to the new layers) |
|
Our reply: Thank you for this note. Now, we modified the presentation of the comparison by adding a new Table in the revised version containing this comparison in the discussion section. Note that we run the algorithms [29] and [30] on the same data. Please see Table 5, Page 12 in the revised manuscript and lines 313-315, page 12.
|
|
2) A statistical analysis of results is missing. The results seem too similar to say there is a real statistically significant difference. |
|
Our reply: Thanks for this valuable note. The p-value is reported in the last paragraph in “Results” section, lines 302-306, page 11.
|
|
3) Since all this new info is extracted from original images using linear operations, I do not understand why the results improve. CNNs are able to extract the features (even in strong nonlinear spaces) so authors must provide an explanation. Is it just because the input tensor is larger so CNN is learning more parameters? In other words, it is because of the network or because of the input signal? Some experiments with simpler architectures can be run and give some light. |
|
Our reply: Thanks for this feedback. The classification capability improvement depends on the CNN structure as well as the input signal. · Adding the thickness maps or reflectivity maps confirms the important features related to the disease to be detected. · Using different deep learning structures and training it on either dataset (gray level images or fused images) proves the enhancement of the accuracy as well as the speed of conversion. · DRNet is simple, more compact, and fast in learning compared to AlexNet, but both of them showed how large the improvements in the accuracy of detection in the fused images dataset compared to the gray images dataset. · In our work, we use interpretable, explainable AI, which means that humans can understand the decision made by AI by importing the important features related to the disease, i.e., reflectivity and thickness. Please refer to: Hagras, Hani. "Toward human-understandable, explainable AI." Computer 51, no. 9 (2018): 28-36. Explainable AI is a new trend that is preferable to doctors and physicians. We have clarified these points in the revised manuscript. Please check lines 233-236, page 9 in the Results section, and lines 323-329, page 12 in the Discussion section. |
|
4) The paper has some typos such as repeated words "the the", "Rreflectivity" and so on. |
|
Our reply: Thanks for these precise notes. We have corrected these errors. Also, we have performed proofreading throughout the revised manuscript to correct other typos.
|
Reviewer 2 Report
A method for DR detection on OCT images has been proposed. The main contribution of the proposed method is to use the thickness channel and the reflectivity channel in addition to the grey-level OCT original image. It is an initial study as the authors themselves described in the paper. It must be extensively revised and submit it in its final form, keeping in view of the following comments.
1. The introduction is smaller than the abstract. It must be revised.
2. The amount of data is too small to train or even to fine-tune a deep CNN model. Only 200 hundred images from 100 images. Two images are from the same patient. What is the impact of only left or only right eye? Is there a difference between left and right eyes? If the images are divided randomly for training, validation and testing, then the images of the same person can be in the training and testing sets. In this case, there is overfitting because of the identity of the person. For experiments, the data must be divided on subject basis i.e. the images of the same person either must be in training or testing set.
3. Keeping in view the scarcity of data, training a CNN model from scratch is not suitable. Overfitting cannot be avoided in this case. It is better to use pre-trained models and fine-tuning. Also, some data augmentation technique must be used even in case of fine-tuning due to the very small amount of data.
4. For experiments, 4-fold CV has been used. It is not good when the amount if available data is very small. At least 10-fold CV must be used. The 10-fold CV experiments can be repeated a number of times and the results must be reported as an average (avg+/-std) over all experiments, not over individual folds. For comparison, some statistical significance test must be used to show that the difference is statistical significant.
5. The results have been reported only using accuracy. Other performance metrics such as sensitivity, specificity and ROC curve must be used for evaluation and comparison.
6. AlexNet has been used; it is an initial, simple but computationally very complex model (has very high number of FLOPs and learnable parameters). The experiments must be performed using advanced CNN models such as RestNet, DenseNet etc.
7. Is the comparison with the SOTA methods on the same data? Please make it clear.
8. In equation 1, why RV and R12 are used for normalization?

Author Response
|
1) The introduction is smaller than the abstract. It must be revised..
|
|
Our reply: Thank you for this valuable note. Now, we concatenate the introduction with the literature review to compose the introduction section, since the literature review can be considered an essential part of the introduction. Please check the expanded Introduction section in the revised manuscript, lines 18-115, pages 1,2, and 3. |
|
2) The amount of data is too small to train or even to fine-tune a deep CNN model. Only 200 hundred images from 100 images. Two images are from the same patient. What is the impact of only left or only right eye? Is there a difference between left and right eyes? If the images are divided randomly for training, validation and testing, then the images of the same person can be in the training and testing sets. In this case, there is overfitting because of the identity of the person. For experiments, the data must be divided on subject basis i.e. the images of the same person either must be in training or testing set. |
|
Our reply: Thanks for this note. We need to confirm the following points: · The images are already divided according to the basis of the same person. All the carried out experiments, in the initial study as well as the currently provided experiments are processed according to this standard. · Of course, there may be a difference between left and right eyes in some subjects, but in most cases, it may be insignificant. Therefore, the images of the same person must be in training or in the testing only. · We mentioned that in the results section “section 3.1”, lines 249-251, page 9.
|
|
3) Keeping in view the scarcity of data, training a CNN model from scratch is not suitable. Overfitting cannot be avoided in this case. It is better to use pre-trained models and fine-tuning. Also, some data augmentation technique must be used even in case of fine-tuning due to the very small amount of data. |
|
Our reply: Thanks for this valuable note. We added other pre-trained models to reinforce the results. In addition, augmentation techniques have been used for the current experiments. We used horizontal flipping and random rotation to increase the data size to 800 images. This is mentioned in section 3.1, lines 251-253, page 9.
|
|
4) For experiments, 4-fold CV has been used. It is not good when the amount if available data is very small. At least 10-fold CV must be used. The 10-fold CV experiments can be repeated a number of times and the results must be reported as an average (avg+/-std) over all experiments, not over individual folds. For comparison, some statistical significance test must be used to show that the difference is statistical significant. |
|
Our reply: Thanks for this valuable note. · We modified all the results presentation. The results of 10-fold are reported exactly as required, Please, check Tables 1, 3, and 4. · The p-value is also reported in the results section, lines 302-306, page 11.
|
|
5) The results have been reported only using accuracy. Other performance metrics such as sensitivity, specificity and ROC curve must be used for evaluation and comparison. |
|
Our reply: Thanks for this valuable advice. We added ROC curves beside the AUC values corresponding to each CNN model used in the results section. Please check figure 8 page 10 and lines 278-282, page 10 in the revised manuscript
|
|
6) AlexNet has been used; it is an initial, simple but computationally very complex model (has very high number of FLOPs and learnable parameters). The experiments must be performed using advanced CNN models such as RestNet, DenseNet etc.
|
|
Our reply: Thank you very much. We added other pre-trained models (ResNet101 and DenseNet) to reinforce and support the proposal. The results are reported in Tables 1, 2, 3, and 4. |
|
7) Is the comparison with the SOTA methods on the same data? Please make it clear.
|
|
Our reply: Some comparison items are with other data and other items are with the same data. It is, now, mentioned in the discussion section and the comparison has clarified this point in Table 5, page 12, and lines 313-315, page 12.
|
|
8) In equation 1, why RV and R12 are used for normalization? |
|
Our reply: We added an explanation by citing suitable references to clarify the usage of the normalization. Please, see lines 160-168, pages 5 and 6, and Refs [33] and [34] in the revised manuscript. · Before estimating the reflectivity, a given OCT B-scan is first normalized by the given equation, as the OCT pixel gray level is not an absolute metric of reflectivity since it depends on some external factors, such as pupil dilation, that affect image quality. For example, the retinal NFL in an eye that is insufficiently dilated may appear darker than in a fully dilated eye, even in the case where both eyes are of the same age and free of pathology. Therefore, a relative metric of reflectivity is used, where the reflectivity of the NFL and other layers is a fraction of the RPE reflectivity. It is standardized with respect to the RPE because that layer is typically preserved in early DR. Please refer to references [33] and [34] in the revised manuscript.
|
Round 2
Reviewer 1 Report
Authors have addressed al my comments. I am still reluctant to believe that the results can be generalized beyond this dataset but at least the experiments run with the available dataset are ok and an statistical analysis carried out.
Add references to CNN pretrained models and Imagenet
Author Response
|
1) Add references to CNN pretrained models and Imagenet |
|
Our reply: Thank you for this valuable. The required references are added, with numbers 35, 36, 37, and 38. Please, refer to their citations at lines 197, 198, and 203. |
Reviewer 2 Report
The authors have given responses to most of my comments. However, they did not address comment 5 completely. All results must be reported in sensitivity (sen+/-std) and specificity (spe+/-std) as well. These are very important metrics in the medical diagnosis domain. Aldo, see some other minor comments on the attached file.

Author Response
|
1) The authors have given responses to most of my comments. However, they did not address comment 5 completely. All results must be reported in sensitivity (sen+/-std) and specificity (spe+/-std) as well. These are very important metrics in the medical diagnosis domain. Also, see some other minor comments on the attached file.
|
|
Our reply: Thank you for this valuable note. We reported all required metrics by adding it to results Tables (Tables 1, 2, 3, and 4). The comments in the pdf- attached file are completely addressed. All modifications are highlighted in the revised manuscript. |